# Consideration of Symptom Management in Advanced Heart Failure

**DOI:** 10.3390/ijerph192215160

**Published:** 2022-11-17

**Authors:** Michael Connolly, Mary Ryder

**Affiliations:** 1School of Nursing, Midwifery & Health Systems, University College Dublin, Belfield, D04 V1W8 Dublin, Ireland; 2Education & Research Centre, Our Lady’s Hospice & Care Services, Harold’s Cross, D6W RY72 Dublin, Ireland

**Keywords:** palliative care, heart failure, chronic obstructive pulmonary disease, symptoms

## Abstract

This work provides an opinion on palliative care for people with heart disease. The work focuses on the management of key symptoms associated with both end stage heart disease, applying a palliative approach and suggesting treatment options. The work also indicates the potential for greater collaboration of specialist teams including specialist palliative care in the assessment and management of patients with complex needs as they progress through their disease trajectory. The practical application of evidence-based guidelines and recommendations is key to the successful individualised management of complex symptoms.

## 1. Introduction

Palliative care is best identified as a set of principles that underpin an approach to care, particularly for individuals with a life-limiting condition. The palliative care approach aims to improve the quality of life of people facing problems associated with life-limiting illness. Palliative care for people with heart disease is often complex from the perspective of healthcare professionals. Palliative care interventions are inclusive of symptom management, psychosocial and spiritual support [1]. The fundamental elements of both chronic illness management and palliative care are symptom management. The approach to symptom management may differ between palliative care and specialist heart failure services. Symptom identification in heart failure is indicative of deterioration and regularly leads to aggressive interventions to alleviate the reported symptom in an efficient manner. Symptom management in palliative care adopts a holistic approach addressing the physical aspect, but also the psychological, social and spiritual aspects of suffering for patients [2]. Therefore, the unpredictable disease trajectory of heart disease, and differentiating between palliative care and specialist palliative care (SPC), and who manages symptoms during the disease trajectory adds to the complexity of choosing optimal care [3]. It is important to recognize that not all of those with advanced heart failure will need referral to specialist palliative care services but those who do should have access to specialist palliative care services to support the management for complex symptoms, particularly at the end stage of disease. The successful management of complex symptoms is best achieved with the provision of evidence-based care [4,5]. Quite often this care is guided by clinical guidelines, however, despite the proliferation of guidelines promoting early access to SPC, the evidence would suggest that they are not regularly used in practice [6,7]. This paper focusses on the consideration of commons symptoms in late stage of heart failure are their management using a palliative care approach. 

## 2. Palliative Care Approach for People with Heart Disease

The overall aim of using a palliative care approach is to promote physical, psychological, social and spiritual well-being and is integral to the clinical care of those with heart failure [8].

Given the complex nature of heart disease, the focus of this section will pay particular attention to palliative care for people with heart failure. It is important to note that Heart Failure is not easily defined given the reality that it is a clinical syndrome characterized by a range of typical symptoms (e.g., breathlessness, ankle swelling and fatigue). Signs can also accompany these symptoms (e.g., elevated jugular venous pressure, pulmonary crackles and peripheral oedema) often caused by a structural and functional cardiac abnormality, resulting in reduced cardiac output and/or elevated intracardiac pressures at rest or during stress. 

It is widely acknowledged by the World Health Organisation and specialist cardiovascular associations internationally that the prevalence of heart failure is soaring. This is largely due to the ageing population, increase in cardiovascular risk factors and improved survival of cardiovascular conditions in the population worldwide. With the inevitable increase in numbers, it is logical to assume that referrals to palliative care for complex symptom management are likely to increase, but only where such services are available. For this reason, it is important that all healthcare professionals apply the palliative care approach when planning care to manage symptoms related to advanced heart disease.

## 3. Managing Common Symptoms

While a range of symptoms accompanies heart disease, a number are common as the disease progresses, including dyspnoea, fatigue and oedema. This discussion will focus on the use of the palliative care approach to manage four specific symptoms; breathlessness, fatigue, anxiety and constipation.

### 3.1. Breathlessness

Breathlessness is the dominant and most frightening symptom for patients with end-stage heart failure. As a symptom, it is difficult to control as the experience for the patient often does not correlate with any measures of its severity or the perceptions of the patient and/or their family.

As a symptom, breathlessness, particularly at rest, is indicative of advancing or worsening disease. The traditional heart failure intervention for the management of breathlessness focuses on the alleviation of fluid overload by increasing diuretics. A more holistic palliative care approach focuses on the individual assessment of needs where it is essential in order to plan interventions, which can either be pharmacological or non-pharmacological.

Opioids remain the medication best placed to reduce the subjective sensation of breathlessness in palliative care. Opiate use is recommended only in selected patients to relieve severe pain or anxiety in patients with heart failure [9]. Although the ideal opioid has not been identified, Morphine is in the main been the drug of choice. Johnson and Currow [10] in a narrative review indicate that there is now moderate level evidence for the use of opioids for breathlessness given as a regular dose rather than as single intermittent doses. When morphine is initiated it should be done at a low dose with increases to the dose being staged slowly until optimal response is reached. Studies indicate that morphine responsiveness to breathlessness occurred in doses of ≤30 mg/24 h orally, with responses seen in a majority at doses of 10 mg/24 h orally, with responses seen in a majority at doses of 10 mg/24 h orally [11,12,13]. A continuing concern when administering opioids for breathlessness is the fear of adverse respiratory effects. However, the evidence appears to suggest that respiratory depression was not observed in the majority of studies [10].

The use of morphine in cardiovascular disease management is associated with acute ST-elevation myocardial infarction [14]. The use of morphine in Cardiovascular medicine is fundamental in the management of pain, not only for comfort effects but because pain is also associated with adverse haemodynamic effects, including increased heart rate, elevated blood pressure and impaired coronary perfusion [14]. However, the administration of morphine for acute decompensated heart failure is an independent predictor of hospital mortality [15]. The evidence has also identified that using morphine with antiplatelet treatment is contraindicated [14]. Whilst this evidence is associated with acute cardiovascular incidence there is limited data exploring the use of opioids in chronic cardiovascular disease. One such study in the United Kingdom [16] has recommended morphine use in advanced heart failure patients to manage breathlessness and concluded that the fears of serious harm such as respiratory depression, excess sleepiness, or cognitive impairment associated with the use of morphine in the patient group are unsubstantiated. The study did not achieve the required sample size for power analysis, however, the research team has recommended further research accompanied by dose titration steps and a side-effect management plan for patients that would improve the safe and effective use of morphine in advanced heart failure [16]. 

### 3.2. Fatigue

Fatigue is a common and often distressing symptom for patients with heart failure. It is also a commonly misunderstood symptom in this patient population, where it is often assumed that the symptom will improve with the removal of fluid-related congestion [17]. While specific screening tool for fatigue in heart failure is available, it is often described by patients as a physical and psychosocial burden, limiting patients’ abilities to perform daily activities, negatively affecting their mood and causing physical pain [18]. Unlike other symptoms, fatigue cannot be treated with pharmacological therapies alone. While fatigue is a common symptom of heart failure and would benefit from palliative care consultation, Feder et al. [6] have suggested that only one in twenty patients with reduced ejection fraction (HFrEF) and one in twenty-five patients with mid-range ejection fraction (HFmEF) and preserved ejection fraction (HFpEF) are likely to receive palliative care. A recent systematic review exploring fatigue in advanced disease reported a dearth of evidence related to diseases other than advanced cancer [19]. This review reported that physical exercise should be considered to reduce fatigue in advanced cancer patients [19], however, this is not a viable option for many advanced cardiac patients. Polikandrioti et al. [20] in their study found that levels of fatigue in heart failure patients ranges from mild to severe, with fatigue higher in hospitalised patients compared to those stable and at home. They also found that patients’ quality of life was affected negatively when they experienced fatigue [21]. Perez-Moreno et al. [20] indicate that between baseline and 6 months worsening fatigue was indicative of worsening outcomes. It is important therefore that patients presenting with advanced heart failure be systematically screened specifically for fatigue so that those at higher risk can be identified and additional supports put in place.

### 3.3. Anxiety

Anxiety increases significantly following diagnosis of heart disease [22]. Such anxiety is often associated with breathlessness. Even when the patient is not experiencing breathlessness, there is potential for residual anxiety and fear regarding the recurrence and likely worsening effect that breathlessness will have as the disease progresses. It is important to note that patients equate the improved quality of life to better psychological well-being than better physical functioning [22]. Palliative care is characterized by its attention to the provision of holistic care, attending to psychological and spiritual needs to the same extent as physician needs. This holistic care is provided by a multidisciplinary team and includes an openness to conversations with patients about their fears and concerns regarding their illness and prognosis, in order to develop a plan of care that attends to the wishes and preferences of the patient and their carers. These conversations will also include a focus on advance care planning, including end-of-life care and preferred place of death.

Treatments can be either psychologically or pharmacologically based depending on the extent of the patient’s anxiety. Psychological therapies can include Cognitive Behavioural Therapy (CBT) and are more likely of benefit at the early stages of the disease combined with instruction on disease self-management [23]. Pharmacological interventions may also be considered and could include antidepressants, which are the main medication used for anxiety, while consideration may also be given to using fewer common agents such as benzodiazepines and antipsychotics. The choice of drug used is dependent on the severity of the symptom.

### 3.4. Constipation

Constipation increases with age and is associated with an increased risk in cardiovascular events. Constipation is a common problem for patients receiving palliative care, particularly those with advanced diseases, both malignant and non-malignant [24]. In palliative care guidance on the assessment and management of constipation in the main comes from an advanced cancer perspective [24]. It is clear that the principles for the management of constipation set out in these guidelines are transferable from cancer care to the area of heart disease where the problem of constipation can be equally evident [25].

The standard clinical definition of chronic constipation, based on “Rome IV Criteria” requires the presence ≥2 of the following symptoms for ≥3 months with ≥6 months: straining during bowel movements; lumpy or hard stool for >25% defecations; sensation of incomplete evacuation for >25% defecations; sensation of anorectal blockage or obstruction for >25% defecations; manual evacuation procedures to remove stool for >25% defecations <3 spontaneous bowel movements per week [26].

As heart disease progresses the continued use of medications, such as anticholinergics, diuretics, and opioids for symptoms that arise are likely to increase the prevalence of constipation. Logically, dehydration is a contributing factor to constipation. Large doses of diuretics are used to treat heart failure fluid overload, however, there is limited research exploring constipation in this population. Holistic palliative care management of constipation is aimed in the first instance at prevention and where constipation occurs to assess and intervene quickly to avoid further complications.

Prevention of constipation forms part of the palliative care plan for the individual with heart disease and should be a focus when prescribing medication that may cause or lead to additional problems with constipation.

A comprehensive assessment is conducted in order to accurately diagnose the presence and potential causes of constipation. A thorough history and physical examination are essential components of this assessment in order to understand the normal bowel routine and ascertain if patients are currently using non-pharmacological (i.e., increased dietary fibre, or over the counter products) or taking prescribed medicinal products for constipation. If prescribed medical products for the management of constipation are being taken these should be reviewed and titrated in the same way as other medications are (Table 1). If the patient is not taking any medical product for constipation, an aperient regime should be introduced if opioids are prescribed for breathlessness. If opioid induced constipation is suspected and confirmed by comprehensive assessment consideration of the use of peripherally acting μ-opioid receptor antagonists may be considered [27].

It is also important to ensure that attention is paid to the provision of optimised toileting ensuring adequate privacy and dignity for patients. 

## 4. Discussion

The evidence indicates that a variety of treatment options for symptom management in advanced heart disease is improving, with the exception of fatigue management. Despite the evidence and recommendations for early SPC interventions and collaboration late referral to specialist palliative care services remains a concern.

The availability and practical application of guidelines are important in order to maximise the supportive aspects of care that palliative care services can provide to patients with end-stage heart disease.

The evidence of the beneficial use of opioids for the management of breathlessness is increasing and challenges the fear of respiratory depression once associated with opioid use. There is however a need for clearer guidance on opiate dose titration when managing breathlessness along with an effective management plan for potential side effects. This can be addressed locally with more collaborative working amongst specialist teams.

There remains a dearth of evidence in respect of the management of fatigue in the non-cancer population. Research on fatigue in advanced non-cancer diseases must be a priority for the future.

Anxiety as a symptom reflects the fears and concerns of patients with end stage disease, particularly the fears associated with recurring and worsening breathlessness. A combined approach to the management of breathless and associated anxiety is recommended, again drawing on the expertise of specialists across disciplines.

While expertise in the management of constipation and opioid-induced constipation resides in specialist palliative care, this symptom is also problematic for the end-stage cardiac disease patient, primarily due to the side effects of prescribed medication. A collaborative approach to managing constipation would be of obvious benefit to the patient.

Future research needs to focus on outcomes of guideline implementation for the management of common end-of-life symptoms, particularly in this population. Research should also focus on the collaborative management of patients with end-stage heart disease by the disease specific specialist supported by specialist palliative care. Individualised person-centred care can only be delivered when evidence-based guidelines and recommendations are implemented and all resources available are used to provide supportive care to patients and their families. Failure to do so, particularly the failure of late referral and indeed not referring at all to specialist palliative care, is a failure to treat and care.

## Figures and Tables

**Table 1 ijerph-19-15160-t001:** Laxative Options.

Laxative	Action	Product
Bulk laxatives	Stimulating	Pysillium, Cellulose
Lubricant laxatives	Soften through penetration	Mineral oil (liquid petrolatum)
Surfactant Detergent	Water and fat absorption	Docusate
Osmotic laxatives	Non-absorbable sugars which assist in retaining fluid in the bowel lumen	Lactulose, polyethylene glycol
Bowel stimulants	Increase colonic motility	Senna, Bisocodyl
Combination	Softener and stimulant	Senna & Docusate

## Data Availability

Not applicable.

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
