# Peer review of "Consideration of Symptom Management in Advanced Heart Failure"

_ijerph, 2022, doi:10.3390/ijerph192215160_

Round 1

Reviewer 1 Report

The authors present a brief report of four symptoms which should be treated in patients with heart failure. The contents are of high interest, unfortunately there are some shortcomings.

In the whole manuscript the authors differ between generalist and specialist palliative care, without clearly describing the differences. Therefore, not every patient with burdensome physical symptoms needs specialist palliative care. As outlined in the Introduction section a more precise definition is desirable.

To my eyes some references are missing e.g. ll 30, 50, 70, 77, 136ff, 149ff, 152, 

Introduction: 

The authors cite the WHO’s definition of palliative care but don’t refer to it (Line 30).

The differentiation between generalist and specialist palliative care is not clearly outlined and remains unclear in the whole manuscript. In line 34 I would like to know, what evidence based care is in that context und would like to be able to read it up in a reference. 

Breathlessness

What about restricting fluids? Refractory dyspnoea is not mentioned. 

It should be pointed out, whether the use of morphine in combination with antiplatelet treatment is contraindicated in the acute or continuous use. 

Fatigue

When systematically screening is mentioned it should be mentioned as well how screening should be performed – and whether there exist recommendations for screening tools?

Consitpation

Does constipation because of the disease itself or because of the disease management? Or because of both?

I guess citation 20 is not adequately cited in context. As far as I know patients without cancer were no included in that review. 

181 missing clamp

Constipation caused by medications is mentioned in l 172. Therefore, I would suggest to also report the OIC and its possible solutions which include PAMORA.

Discussion

The authors talk about opioids in the whole manuscript. Is there a reason for changing to opiats in the discussion?

Author Response

Response to Reviewer1 comments in attached file.

Reviewer 2 Report

In their article, the authors present recommendations for symptom management in palliative heart failure care. As presented by the authors, there is a huge gab between the need for palliative HF care and its availability. Symptom management is a cornerstone of HF care especially in a palliative care situation and physicians should be aware about adequate treatments. The topic of the paper is therefore of interest to many physicians. However, the paper's novelty is limited since there are already many studies (including small RCTs) and reports on this topic. 

A quick PubMed literature search on "heart failure" and "palliative care" (restricted to titles or abstracts) yielded more than 1000 results, including reviews, systematic reviews, meta-analyses and small RCTs. While the introduction in its current version highlights the need for palliative symptom management (which I agree with), it does not adequately present the current evidence on the topic.

Also, the authors do not describe any kind of methodolgy. To me, it is not clear whether the manuscript presents an expert opinion statement or some kind of review. In case it is a review, the methodolgy should be described. In case it's an expert opinion, this should clearly be stated in the abstract and the introduction section.

Breathlessness is clearly the most debilitating symptom in advanced HF. In the manuscript, the authors focus on the use of morphines to relieve dyspnoea. However, there are many other treatment options that are not mentioned in the paper. Even in a palliative setting, cardiac resynchronisation therapy (without implantation of a defibrillator)  may improve HF symptoms. Also, the interventional reduction of atrioventricular valve regurgitation may contribute to symptom relief. In case of fluid retention despite high dose diuretics, ultrafiltration therapies may improve fluid balance and dyspnoea. I would also be interested to receive adivce with regard to deprescribing of HF medications with regard to symptom management. 

The authors state that physical exercise may not be a treatment option in advanced HF patients. I disagree! There are many studies showing that even the most advanced HF patients (NYHA IIIb-IVa) benefit from physical exercise. In my opinion, physical extercise should only be avoided in acute decomensated HF. Please discuss!

The authors write: "Pharmacological interventions may also be considered and could include antidepressants, which are the main medication used for anxiety, while consideration may also be given to using less common agents such benzodiazepines and antipsychotics. The choice of drug used is dependent on the severity of the symptom." My comments:

- Tricyclic antidepressants should be avoided in HF patients because they increase the risk of malignant arrythmia and myocardial infarction. Other antidepressants should be used with caution and regular checks of QT intervals.

- SSRIs have not proven any benefit in patients with symptomatic HF. Please discuss!

What about other symptoms in patients with advanced HF, such as oedema/ ascites or dizziness?

Please provide information on author contributions, funding, data availability and conflicts of interest

Author Response

Response to Reviewer 2 comments in attached file

Round 2

Reviewer 1 Report

Thank you for reviewing the manuscript, which provides a good overview as a Brief Report.